# Molecular identification of the wheat male fertility gene *Ms1* and its prospects for hybrid breeding

Elise J. Tucker[1], Ute Baumann[1], Allan Kouidri[1], Radoslaw Suchecki [1], Mathieu Baes[1], Melissa Garcia[1], Takashi Okada[1], Chongmei Dong[2], Yongzhong Wu[3], Ajay Sandhu[3], Manjit Singh[3], Peter Langridge[1], Petra Wolters[3], Marc C. Albertsen[3], A. Mark Cigan[3] & Ryan Whitford [1]

The current rate of yield gain in crops is insufficient to meet the predicted demands. Capturing the yield boost from heterosis is one of the few technologies that offers rapid gain. Hybrids are widely used for cereals, maize and rice, but it has been a challenge to develop a viable hybrid system for bread wheat due to the wheat genome complexity, which is both large and hexaploid. Wheat is our most widely grown crop providing 20% of the calories for humans. Here, we describe the identification of *Ms1*, a gene proposed for use in large-scale, low-cost production of male-sterile (*ms*) female lines necessary for hybrid wheat seed production. We show that *Ms1* completely restores fertility to *ms1d*, and encodes a glycosylphosphatidylinositol-anchored lipid transfer protein, necessary for pollen exine development. This represents a key step towards developing a robust hybridization platform in wheat.

[1] Australian Centre for Plant Functional Genomics, School of Agriculture, Food & Wine, University of Adelaide, Waite Campus, Urrbrae, SA 5064, Australia. [2] Plant Breeding Institute, University of Sydney, PMB 4011, Narellan, NSW 2567, Australia. [3] DuPont Pioneer Hi-Bred International Inc., 7250 NW 62nd Avenue, Johnston, IA 50131-0552, USA. Elise J. Tucker and Ute Baumann contributed equally to this work. Correspondence and requests for materials should be addressed to R.W. (email: Ryan.Whitford@adelaide.edu.au)

With the predicted growth in world population to over nine billion by 2050, the Food and Agriculture Organization of the United Nations (July 2005) set a target of 60% increased food production by that year. This is an ambitious target for two reasons: there are serious concerns about the viability of existing production systems and the sustainability of current growth rates in crop production, and the predicted environmental changes are expected to have an overall negative effect on agricultural production, with serious crop declines in some countries. Wheat is grown more widely than any other crop and delivers around 20% of our food calories and protein[1]. To increase global production by 60% will require a lift in the rates of gain from the current 1 to 1.6% per annum. Improvements in disease resistance and stress tolerance offer opportunities for small increases in productivity but major jumps in yield are hard to achieve and are expected to come through shifts in the way we breed wheat and other crops. However, many important new genetic and genomic technologies are difficult to apply to wheat since this plant has a large complex genome, an allohexaploid, which is 50 times larger than rice.

One of the most promising options for achieving significant boosts in yield across diverse production environments is through hybrid breeding. Hybrids offer two important advantages: first, heterotic yield gains of well over 10%, and improved yield stability have been reported[2–4], and second, hybrid seed production would act as a major stimulant for investment in wheat improvement from both the public and private sectors. However, the competitiveness of wheat hybrids relative to line varieties will depend on hybrid seed production costs[5].

Lowering hybrid seed production costs depends on a reliable and inexpensive system that forces outcrossing. Wheat male sterility and restoration systems were first developed in the 1960s, but many of them were proved to be impractical and deemed commercially high risk[6]. Relative to systems based on chemical hybridizing agents and cytoplasmic male sterility, the use of non-conditional nuclear-encoded recessive male steriles (ms) would offer major advantages for hybrid breeding. The value of recessive male steriles was first recognised in 1972 with the proposal of the XYZ system[7]. This system aimed to overcome the costs associated with propagating pure stands of male steriles by cytogenetic chromosomal manipulation[7, 8]. A further advantage of recessive male steriles came through the opportunity to broaden parental line choice, avoid negative alloplasmic and cytoplasmic yield penalties, as well as alleviate the problems associated with incomplete fertility restoration. A cost-effective and flexible hybridization platform that uses a recessive male sterile is Seed Production Technology (SPT)[9] developed for maize and rice hybrid seed production (Supplementary Fig. 1). This platform overcomes many of the problems with large-scale production of male steriles for use as female parents in hybrid breeding. SPT uses a maintainer line solely for the propagation of non-GM homozygous recessive male-sterile parents; therefore, F$_1$ hybrids provided to farmers are considered to be non-GM.

Developing an equivalent platform for hybrid wheat breeding requires the identification of a suitable non-conditional, nuclear-encoded recessive male sterile. These types of mutants are particularly rare and difficult to detect in polyploids due to genetic redundancy. Many of our major crops and food plants are polyploids, including wheat, oats, potato, sweet potato, peanut, sugarcane, cotton, kiwifruit, strawberry, and plums. For example, only ten nuclear-encoded wheat male-sterile mutants have been identified to date[6], in contrast to 108 mutants in diploid barley[10, 11]. Polyploidy not only makes it difficult to find suitable male-sterile mutations but also complicates deploying mutants since multiple mutations would be needed to deal with genetic redundancy[12] and this increases breeding costs and population sizes needed for introgression of each additional mutation. The most cost-effective mutants would be single locus encoded. In wheat, only two of the ten mutant loci are reported to fit this criterion. These are ms1 and ms5 located on chromosomes 4BS and 3AL, respectively[13].

The first ms1 mutant was observed in Australia in the late 1950s[14]. This spontaneous mutant named Pugsley's male sterile was followed by the identification of Probus and Cornerstone male steriles from an X-ray-induced wheat mutant population[15, 16]. Cytogenetic and linkage analysis showed these to be allelic and they were designated as ms1a, ms1b and ms1c, respectively[15, 17–19]. In 1976, additional monogenic recessive male steriles were identified from an ethyl methanesulfonate (EMS)-treated population[20]. Three mutants were allelic to ms1, and designated as ms1d, ms1e and ms1f[13, 21] while the fourth mutant was nonallelic to ms1 and designated as ms5[13]. However, even for ms1, the variability between backgrounds and mutant alleles, and problems with male sterility penetrance were reported[22–24]. In order to address these problems, it is necessary to identify the gene underlying the Ms1 locus and explain its function.

Here, we describe the identification of the Ms1 gene sequence (TaMs1) by map-based cloning and demonstrate its function in male fertility by complementation of the ms1d mutant. TaMs1 encodes a glycosylphosphatidylinositol (GPI)-anchored lipid transfer protein, which is necessary for pollen exine development. The identification of the Ms1 gene sequence represents a key step towards developing a robust hybridization platform in wheat similar to the maize SPT.

## Results

**Ms1 encodes a GPI-anchored lipid transfer protein**. We followed a map-based cloning approach to isolate the Ms1 gene sequence (chr. 4BS). Using syntenic regions on chromosomes 1, 3 and 4 from rice, Brachypodium and barley, respectively, we generated markers and tested their presence or absence in male-sterile mutants ms1a, ms1b and ms1c. The results revealed that ms1a and ms1c are terminal deletions while ms1b is an interstitial deletion of the chromosome 4BS covering approximately 14 centimorgans (cM) (Fig. 1). Ms1-flanking markers were identified by their presence in ms1b and their absence from ms1c. Using a population representing 7000 meioses and segregating for ms1d, we delimited the Ms1 locus to a 0.5 cM interval between markers ×27140346 and ×12360198. Probes designed within the region bounded by these markers, were used to isolate and sequence BACs from durum and hexaploid wheat. Marker development from BAC-derived sequences and analysis of 14 recombinants across the region, further delimited Ms1 between markers 007.033.1 and 007.0046.1. The mapped 251- Kb interval contains eight intact genes and one likely pseudogene (Supplementary Table 1).

RNAseq-based expression profiling identified one of these eight genes to be preferentially expressed in floral tissues (Supplementary Table 1). This gene (TaMs1) is predicted to encode a 219 amino acid polypeptide with a similarity to a large family of GPI-anchored lipid transfer proteins (LTPGs), for which it is a member of a Poaceae-specific clade (Supplementary Fig. 2). This gene was confirmed as Ms1 through in planta complementation of the ms1d mutation and identification of the causative lesions in ms1d, ms1e, ms1f and a newly identified TILLING mutant (described below).

**TaMs1 is necessary for pollen exine integrity**. Arabidopsis harbours over 20 LTPG genes for which only two of them have been characterised[25, 26]. AtLTPG1 and AtLTPG2 are required for cuticular wax accumulation or for export onto stem and silique

surfaces. Epicuticular wax has lipid precursors common to sporopollenin, the major constituent of pollen exine, which is produced in sporophytic tapetal cells and transported to developing microspores in structures called orbicules. The analysis of *ms1* anthers revealed a disrupted orbicule and a pollen exine structure (Fig. 2a), which was first observed in early uninucleate microspores and typified by ectopic exine deposition and reduced electron-dense materials at the tapetal cell surface (Supplementary Figs. 3 and 4). No differences in the surface cuticle layer were observed between *ms1* and wild-type anthers (Supplementary Fig. 5). Furthermore, metabolomic profiling revealed that *ms1* anthers accumulate lipid monomers of sporopollenin (C16 and C18 long-chain fatty acids) relative to the wild type (Supplementary Fig. 6). Taken together, this suggests that *Ms1* is necessary for sporopollenin biosynthesis or transport. Transcriptional β-glucuronidase fusions and homeologue-specific qRT-PCR revealed only the B-genome-derived *TaMs1* is to be expressed during early microspore development (Fig. 2b, c).

**TaMs1 exhibits functional divergence from its homeologues.** Since no obvious differences in *TaMs1* coding potential were detected between homeoloci, the basis for *TaMs1*'s subfunctionalization between homeoloci is likely to be due to variation in transcription. We suggest that functional homeoalleles may still exist in the cultivated germplasm pool and this could account for the reports of poor sterility penetrance dependent upon the genotype and the mutant allele[22–24]. In each of these cases, the loss of *Ms1* is either via a large deletion (*ms1a* and *ms1c*) or chromosome arm (DT4BL) replacement; therefore, restoration of fertility is unlikely to be a consequence of the B-genome-derived *TaMs1*. By performing a *TaMs1* homeologue-specific qRT-PCR on anthers isolated from a partially fertile homozygote for *ms1c*, we attempted to answer the question on whether *TaMs1* homeoalleles can transcriptionally compensate for the loss of *TaMs1*-B. However, this does not seem to be the case since *TaMs1* and homeologous transcripts from cv. Cornerstone were all below the detection limits (Supplementary Fig. 7). It is, therefore, possible that other genomic loci are associated with fertility restoration in lines carrying *ms1* deletions.

The variable penetrance of *ms1* large-deletion mutants led us to investigate the utility of the available *Ms1* mutant alleles derived from an EMS-treated population[27]. Chromosome 4BS-specific full-length coding sequences from wild-type (*Ms1*) fertiles and the EMS-derived *ms1d*, *ms1e* and *ms1f* steriles were isolated and sequenced. A comparison of *Ms1* to *ms1*-derived sequences identified unique single-nucleotide transitions for each mutant allele (Fig. 3a). Transition G329A is unique to *ms1d* and unlikely to be a natural allelic variant, considering that the wild-type G is detected in all 192 spring wheat varieties tested (Supplementary

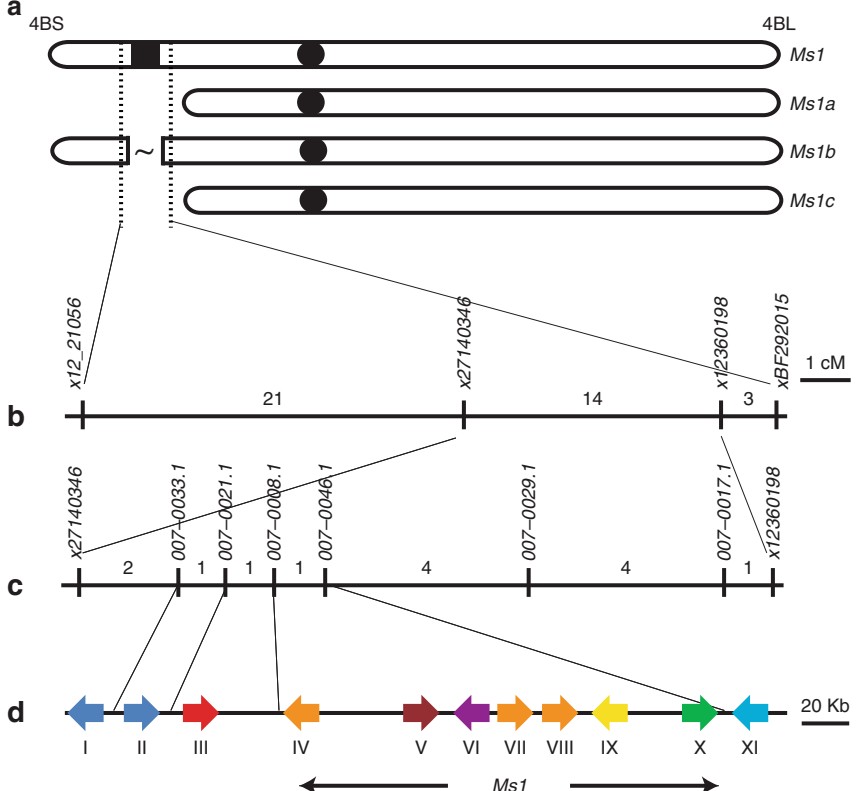

**Fig. 1** Map-based cloning of the *Male sterility 1* locus on chromosome 4BS. *Ms1* was initially mapped to the interval between × *12_21056* and *xBF292015* based on genotyping a (**a**) deletion mutant allele series, and then to an (**b**) ~ 0.5 cM interval between ×*27140346* and ×*12360198* based on 7000 F₂ segregants. **c** Fine mapping using 14 recombinants delimited *Ms1* to a (**d**) 251-Kb genomic region in wheat. Marker names are in italics. The numbers indicate recombinants identified for each marker interval. Coloured arrows I to XI denote the position and orientation of predicted wheat genes with a similarity to *Brachypodium* genes Bradi1g13040 (Cupin domain-containing protein), Bradi1g13040 (Cupin domain-containing protein), Bradi2g05445 (60S ribosomal protein), Bradi1g13030 (Lipid Transfer Protein-Like 94), Bradi4g44760 (F-box/LRR-repeat protein 3), Bradi1g69240 (U-box domain-containing protein), Bradi1g13000 (Lipid Transfer Protein-Like 72), Bradi1g12990 (Lipid Transfer Protein-Like 71), Bradi1g12980 (Putative Parafibromin), Bradi1g12970 (Putative GNAT family acetyltransferase) and Bradi1g12960 (DUF581 domain-containing protein), respectively. The sequence is available via NCBI GenBank accession code KX447407

Table 2). This SNP, at the first *Ms1* exon–intron boundary induces a cryptic splice site in the first intron, resulting in a coding sequence frame shift (Fig. 3b). *ms1e* transition C1435T is coupled with a 1 bp deletion in the second exon, resulting in a frame shift prior to the predicted C-terminal GPI anchor domain. *ms1f* transition G155A changes a highly conserved cysteine residue to a tyrosine (C52Y). This site is one of the eight cysteine motifs characteristic of LTP domains that appear to be important for the structural scaffold necessary for lipid binding[28]. In a parallel approach, we identified a G178A transition (designated as *ms1h*) in a TILLING screen of a soft wheat cultivar *QAL2000*, that induced male sterility similar to other mutant alleles (Supplementary Fig. 8) when it was in the homozygous condition (Fig. 3a)[29]. This mutation changes an aspartic acid to an asparagine (D60N) within the conserved LTP domain. Taken together, these findings indicate that both the GPI anchor and putative lipid-binding domains are necessary for Ms1 functionality.

**TaMs1 functionally complements *ms1d*.** The SPT hybridization platform incorporates a maintainer line (Supplementary Fig. 1) capable of propagating non-GM nuclear male-sterile lines for use as female parents in hybrid production. The SPT maintainer line is a homozygous recessive male sterile transformed with a SPT construct containing (i) a complementary wild-type male fertility gene to restore fertility, (ii) an α-amylase gene to disrupt pollination and (iii) a seed colour marker gene. We demonstrated that the α-amylase gene and seed colour marker function in wheat (Supplementary Fig. 9). However, the remaining key component of the SPT hybridization platform requires the demonstration of complementation of male sterility to fully restore fertility[9]. Therefore, we tested the ability of this gene sequence to complement *ms1d*. A 4.4-Kb genomic fragment containing *Ms1* was synthesised (*TaMs1*) and introduced into the wheat cultivar *Gladius* segregating for *ms1d*. Eleven independent *Agrobacterium*-mediated T₀ transformants were generated (Supplementary Table 3). Four SNP markers closely flanking the *Ms1* locus allowed the selection of T₀ regenerants that were homozygous for *ms1d*, whilst a seed-screenable marker (*DsRed*) was used to confirm the presence of *TaMs1*. SNP and seed colour detection identified six homozygous *ms1d* T₀ regenerants with the introduction of the *TaMs1* gene. Selfed seed set analysis showed that all six T₀ regenerants were fully fertile (Fig. 3c, Supplementary Table 3). Seventeen T₁ progenies for two independent T₀ lines (Event 1 and Event 7) were assayed for both the copy number and zygosity of the introduced *TaMs1* (Supplementary Table 4). The results revealed that all progenies were homozygous for *ms1d* with either zero, one or two copies of the exogenous *TaMs1*. Those progenies with no detectable introduced *TaMs1* were male sterile whilst those containing either one or two copies of the *TaMs1* transgene were self-fertile. These findings demonstrate that *Ms1* can fully restore fertility to the homozygous *ms1d* mutant.

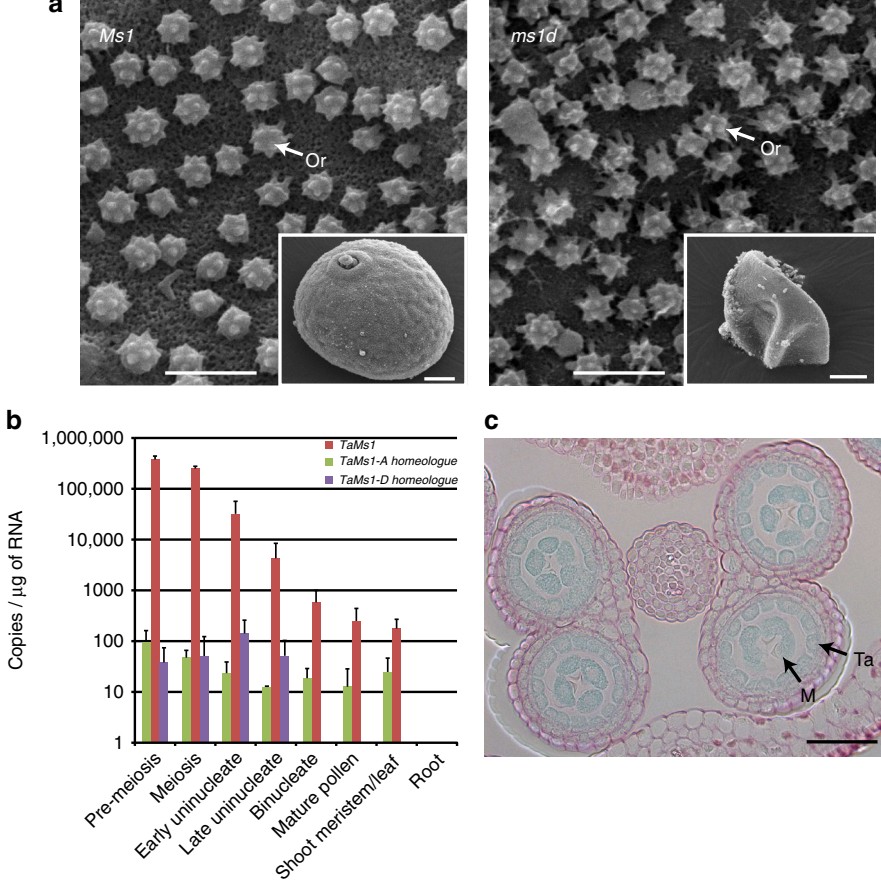

**Fig. 2** *TaMs1* confers male fertility and is expressed in developing wheat anthers. **a** Scanning electron micrographs showing the defects in tapetal cell surface-localised orbicule (Or) structures and pollen coat (inset) within male sterile (*ms1d*) vs. wild-type anthers (*Ms1*). Scale bars: 2 μm (inset 10 μm). **b** *TaMs1* and homeologue mRNA levels as detected by qRT-PCR in premeiotic to mature pollen-containing anthers, leaf/shoot apical meristem and roots. The data are means ± s.e.m (n = 3 biological replicates). **c** Histochemical GUS analysis of wheat anthers expressing a translational GUS fusion with *TaMs1* (native promoter). Transverse section of a wheat anther containing microspores undergoing meiosis showing cell-type-specific GUS expression. Ruthenium red-stained cell walls (pink). Scale bar: 50 μm. (Ta Tapetal cell, M Microspore)

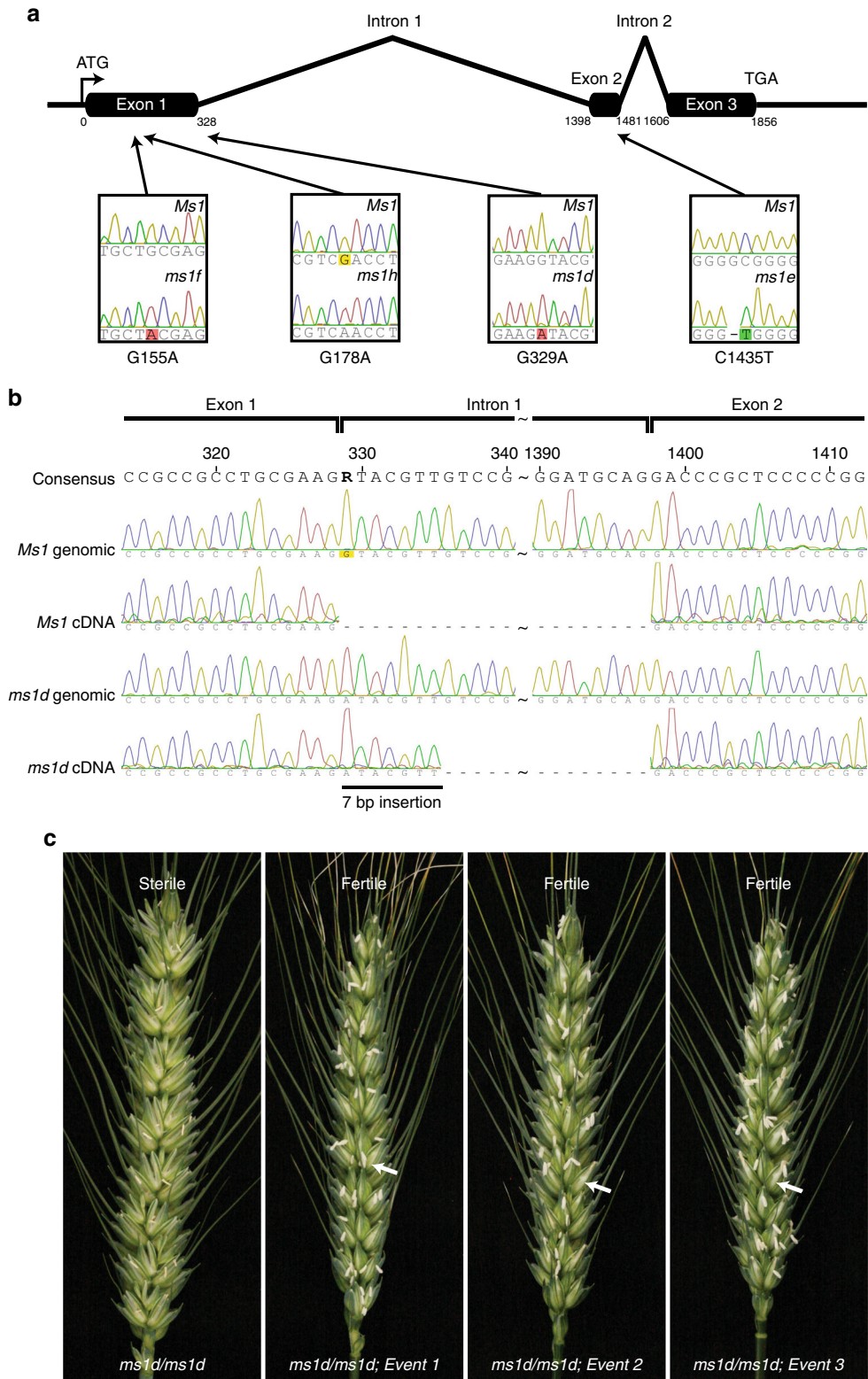

**Fig. 3** Identity of male sterility inducing lesions within *TaMs1* and in vivo complementation of *ms1d/ms1d* by *TaMs1*. **a** Schematic representation of the *TaMs1* (the three exons are shown as black boxes) gene depicting the relative positions (indicated by solid lines with arrowheads) of EMS-derived lesions (chromatogram insets) for *ms1d, ms1e, ms1f* and *ms1h*. Polymorphisms cause either a frame shift (*ms1d, ms1e*) or an amino acid transition in a conserved residue of TaMs1 (*ms1f, ms1h*). **b** Sequence chromatogram comparison between both mutant (*ms1d*) and wild-type genomic cDNAs. Polymorphism G329A of *ms1d* induces the use of a cryptic splice site (highlighted by a 7 bp insertion) within *TaMs1*. **c** Stable complementation of the *ms1d* mutant by *TaMs1*. Mature inflorescences of male-sterile *ms1d/ms1d*, and three independent transformants (Events 1–3), each homozygous for *ms1d*, and showing a self-seed set (arrow). Scale bars: 1 cm

## Discussion

An ambitious target of a 60% increase in wheat production by 2050 has been set to meet the predicted demand for this crop (FAO). A viable hybrid seed production system could deliver a third or more of this gain and currently, it is difficult to see any other technology that could achieve an equivalent impact. Several components are needed for an efficient hybrid system, which include optimising hybrid gains through defining heterotic pools and improving pollen vectoring by transitioning from a self-pollinating flower structure to the one that facilitates outcrossing. Based on experience in other hybrid systems, these components can be generated through targeted selection. However, forcing outcrossing through a reliable and manageable male sterility system requires the development of systems that are able to cope with the complexity of the hexaploid wheat genome.

The use of recessive male sterility has been an attractive prospect for hybrid seed production in wheat since it was first proposed in the 1950s but translating the concept to reality has proved to be elusive. The conundrum has been to maintain the male-sterile lines for use as pollen acceptors in commercial hybrid seed production fields. For example, the cytogenetic 4E-*ms* system, which utilises the novel mutant *ms1g* allele and a fertility-restoring chromosome from *Agropyron elongatum* ssp. *ruthenicum* Beldie (4E)[30] was initially reported to be successful in enabling the production of a male-sterile seed. However, reports have since indicated residual pollen transmissibility of chromosome 4E,[31] resulting in a selfed seed set within the male-sterile stands and thus reducing the purity of the hybrid seed produced. Therefore, alternate approaches are being sought to enhance this system. Now, with the isolation of *Ms1* and characterisation of most known mutations at this locus, we finally have the tools needed to develop an efficient means of bulking a male-sterile female-inbred seed (*ms1/ms1*), as is the case for SPT in maize. This represents a major cost saving for hybrid seed production and can overcome the seed purity issues inherent to the 4E-*ms* system. Coupled with a functional α-amylase gene for wheat pollen disruption and a seed-selectable marker, the identity and ability of the *TaMs1* gene sequence to completely restore viable pollen production in *ms1* plants represents the last critical step towards developing SPT for wheat.

To our knowledge, this study is the first to report a role of a GPI-anchored LTP in pollen exine development[32]. Given the observation that *ms1d* anthers contain pollen deficient in exine structural integrity, and that metabolomic profiling revealed a significant accumulation in free C16 and C18 fatty acids, it is reasonable to expect this to be a consequence of disruption in sporopollenin precursor transport from sporophytic tapetal cells to developing microspores. The finding that anther epidermal wax layers were not disrupted in *ms1d* anthers also indicates that TaMs1's role is specific to exine development. Further studies are needed to elucidate the subcellular localisation of TaMs1 and its direct involvement in lipid binding.

*TaMs1*'s identification now also raises the question on whether previously reported male-sterile mutants may represent additional mutant alleles of this gene or its homologues. Apart from the dominant male-sterile mutants *Ms2*[33, 34] and *Ms4*[35], no other mutants have been reported on homeogroup four. Both *Ms2* and *Ms4* were reported to be located on chromosome 4DS. However, recent cloning and complementation of the *Ms2* gene sequence *WMS*, reveals that it does not encode a *TaMs1* homologue, but represents a novel gene sequence expressed only in sterile anthers as a consequence of a transposon insertion[36, 37]. Further research is necessary to determine whether *Ms4* is a mutant of the *TaMs1*-D homeologous gene sequence.

From a traditional breeding perspective, the molecular identity of the *TaMs1* gene sequence now allows the development of germplasm-specific markers for fast-tracking *ms1* introgression into diverse female-inbred parental lines. Moreover, complementation studies demonstrate that *ms1* is unique and contrasts with other reported wheat *ms* mutant alleles[38] in that *ms1* behaves as a single-mutant locus in hexaploid wheat and a single copy of *Ms1* restores fertility. Given the characterisation of the *ms1* alleles described here and the observed variation in penetrance of sterility between these different alleles, understanding the relationship of these mutations to pollen production, as well as optimising *ms1* as a system for the production of a hybrid seed is now possible through the adoption of advanced breeding technologies such as gene editing[39, 40]. Gene editing would enable the generation of novel *ms1* alleles and allow simultaneous testing in different isogenic wheat backgrounds (e.g. spring and winter or hard and soft wheat). Further, once highly penetrant *ms1* alleles are identified, rather than introgression through conventional backcrossing, this new variant allele could be rapidly introduced into the most elite genetics by directly editing *TaMs1*. Adoption of new breeding technologies is likely to be particularly valuable in translating the results presented here in wheat to other polyploids suffering from a similar paucity of mutants. Such new technologies are necessary for the development of hybridisation systems to exploit heterosis as a means to increase seed production as global population increases.

## Methods

**Plant and DNA materials.** Bread wheat lines used for fertility phenotyping, molecular marker development and genetic mapping were cv. Chinese Spring, Chinese Spring-derived nullisomic tetrasomic stocks[41] (N4AT4D, N4AT4B, N4BT4A, N4BT4D, N4DT4A and N4DT4B), cv. *Gladius*, cv. Pugsley's male sterile (*ms1a*)[14], cv. *Probus* (*ms1b*)[16], cv. *Cornerstone* (*ms1c*)[18], cv. *Chris* and *Chris*-EMS-mutagenised lines FS2 (*ms1d*), FS3 (*ms1e*) and FS24 (*ms1f*)[21]. All cultivars, breeders' lines or DNA samples for marker screening were obtained from various Australian wheat-breeding programmes, Australian Wheat and Barley Molecular Marker Program or the Australian Winter Cereals Collection (AWCC). The BAC library was derived from the *Triticum turgidum* ssp. durum cv. Langdon[42]. The EMS-mutagenised population used for TILLING was derived from the soft bread wheat cv. QAL2000[29].

**Plant growth and phenotyping.** Plants for genetic mapping, cytological examination, expression analysis, TILLING and transformation donor material were sown at 5–6 plants per 6 l (8-inches diameter) pot containing soil mix. The soil mix consisted of 75% (v v$^{-1}$) Coco Peat, 25% (v v$^{-1}$) nursery-cutting sand (sharp), 750 mg l$^{-1}$ CaSO$_4$.2H$_2$O (gypsum) 750 mg l$^{-1}$ Ca(H$_2$PO$_4$)$_2$.H$_2$O (superphosphate), 1.9 g l$^{-1}$ FeSO$_4$, 125 mg l$^{-1}$ FeEDTA, 1.9 g l$^{-1}$ Ca(NO$_3$)$_2$, 750 mg l$^{-1}$ Scotts Micromax micronutrients and 2.5 g l$^{-1}$ Osmocote Plus slow-release fertilizer (16:3:9) (Scotts Australia Pty. Ltd.). The pH was adjusted between 6.0 and 6.5 using two parts of agricultural lime to one part of hydrated lime. Potted plants were grown either in controlled environment growth rooms at 23 °C (day) and 16 °C (night) or similarly temperature-moderated glasshouses in which the photoperiod was extended using 400 W high-pressure sodium lamps in combination with metal halide lamps to 12 h over winter months.

Individual plants were assessed for self-fertility by placing and sealing a glassine bag over each head before anthesis. Between three and ten heads per plant were collected for seed counting. The two basal and two apical sspikelets per head were eliminated from analysis due to their incomplete development. The total seed set and numbers of florets were counted on a per-head basis. The percentage of fertility for each spike or plant was calculated as follows:

$$\% \text{ fertility} = \frac{\text{Total number of seeds per spike}}{\text{Total number of 1°, 2° and 3° florets per spike}} \times 100$$

A plant was deemed to be self-fertile if the total calculated percentage of fertility was greater than 60% or was equivalent to a wild-type control.

Pollen viability was assessed for three isolated anthers per plant (n = 3) by either acetocarmine–glycerin or Lugol (1% I$_3$K solution) staining. Dissected anthers were mounted on glass microscope slides and pollen grains (n > 500 per sample) counted for staining by visualising them on a Zeiss Axio Imager M2 optical microscope coupled with a CCD camera (The University of Adelaide microscopy). Stainable pollen is represented as the mean ± standard deviation for nine samples per genotype.

**Histochemical staining cytological examination.** Anthers containing premeiotic microspores to mature pollen were isolated from wheat plants identified to contain the *TaMs1::gusplus* cassette. Histochemical GUS activity was detected using 5-

bromo-4-chloro-3-indolyl-beta-D-glucuronic acid (Gold Biotechnology, Inc). The samples were incubated in a 1 mM X-Gluc solution in 100 mM sodium phosphate at pH 7.0, 10 mM sodium ethylenediaminetetraacetate, 2 mM FeK$_3$(CN)$_6$, 2 mM K$_4$Fe(CN)$_6$ and 0.1% Triton X-100. After vacuum infiltration at 2600 Pa for 20 min, the samples were incubated overnight at 37 °C. Anther samples were then immersed in a fixative solution of 4% sucrose, 1× PBS, 4% paraformaldehyde and 0.25% glutaraldehyde, at 4 °C overnight. The samples were then dehydrated in ethanol series of increasing concentrations (30, 50, 70, 85, 90, 95 and 100%). Tissues were embedded in Technovit® resin, then sectioned on a microtome to a thickness of 8–14 μm, counterstained with ruthenium red and DPX mounted (Sigma, St. Louis, MO) on glass slides. The sections were observed using a LEICA ASLMD laser dissection microscope coupled with a CCD camera (The University of Adelaide microscopy).

**Electron and light microscopy.** Sterile (*ms1d*) and fertile (*Ms1*) mature anthers before dehiscence were fixed with either 4% paraformaldehyde, 1.25% glutaraldehyde and 4% sucrose in phosphate-buffered saline (PBS) at pH 7.4, for 16 h at 4 °C for scanning electron microscopy (SEM) or 3% glutaraldehyde in 0.1 M phosphate buffer at pH 7.0 overnight for transmission electron (TEM) or light microscopy. The samples for SEM were rinsed twice with PBS at pH 7.4 for 5 min, whereas the samples for TEM and light microscopy were washed twice with 1 x PBS and embedded in 2% low melting point agarose (Sigma, St. Louis, MO) in 1 x PBS for sample orientation and sectioning, and then dehydrated using a series of graded ethanol solutions (30%, 50%, 70%, 85%, 90% and 95%) each for 60 min. The samples were then infiltrated 3 times, each for 60 min, in 100% ethanol. The samples were either embedded in LR white resin, sectioned (2 μm) and stained with 0.05% toluidine blue stain and mounted on slides in DPX solution (Sigma, St. Louis, MO) for light microscopy or dissected, then they were critical point dried and sputter coated with platinum (BalTec CPD030 Critical Point Dryer) for SEM. 70–80 nm ultrathin anther sections were prepared and stained in 4% uranyl acetate followed by Reynold's lead citrate (The University of Adelaide microscopy)[43]. SEM and image capture was performed at an accelerating voltage of 10 kV (Philips XL20 SEM w EDAX EDS), whereas TEM and image capture was performed on a Phillips CM-1000 TEM (The University of Adelaide microscopy). Light microscopy images were captured using a Zeiss Axio Imager M2 optical microscope (Zeiss, Germany).

**Fatty acid profiling.** Approximately 50 frozen anthers were transferred into pre-chilled cryogenic mill tubes and weighed accurately. A 300 μl aliquot of 1:3:1 chloroform:methanol:water containing a 30 μM internal standard ($^{13}$C$_1$ myristic acid) was added to each sample tube. Dried samples and a fatty acid calibration mix (Supelco®37 Component FAME Mix) was prepared by adding 25 μl of 2:1 chloroform:methanol followed by shaking at 37 °C for 30 min. The samples were then derivatised using 5 μl of Meth-Prep™ II (Grace Davison Discovery). 1 μl was injected onto the GC column. The GC-MS apparatus comprised of a Gerstel 2.5.2 autosampler, a 7890A Agilent gas chromatograph and a 5975C Agilent quadrupole mass spectrometer (Agilent, Santa Clara, USA). The mass spectrometer was calibrated according to the manufacturer's recommendations using *tris*-(per-fluorobutyl)-amine (CF43).

Gas chromatography was performed on a VF-5ms column (Agilent Technologies, Australia). The injection temperature was set at 250 °C, with the MS transfer line at 280 °C, the ion source adjusted to 250 °C and the quadrupole at 150 °C. Helium was used as the carrier gas at a flow rate of 1.1 ml min$^{-1}$. The corresponding GC-MS method was performed using the following temperature programme; start at an injection of 50 °C, hold for 1 min, followed by a 15 °C min$^{-1}$ oven temperature ramp to 230 °C; hold for 3 min, followed by a 10 °C ramp to 300 °C.

Mass spectra were recorded at 2 scans s$^{-1}$ with an *m/z* value of 50–600 scanning range. Both chromatograms and mass spectra were evaluated using the MassHunter Workstation software version B.07.00 (Agilent, Santa Clara, USA). The retention times and mass spectra (unique qualifier ions) were identified and compared directly to standards from a commercially available fatty acid methyl ester mix (Supelco®37 Component FAME Mix, 47885-U, Sigma-Aldrich). All fatty acid methyl esters identified were quantified using prepared calibration curves from the stock Supelco®37 Component FAME Mix in the linear range from 2.5 to 150 μM for each lipid class.

**Mapping Ms1.** Using sequence collinearity among chromosomes 1, 3 and 4 from rice, *Brachypodium* and barley[44], respectively, we were able to identify gene sequences from these species and use them in a BLASTn search of ESTs and homeogroup four-derived genomic survey sequences from the bread wheat cv. Chinese Spring[45]. Chromosome arm 4AL, 4BS and 4DS-derived genomic sequence contigs were then used to develop PCR-based markers that were subsequently validated for sub-genome specificity by amplification on the Chinese Spring-derived nullisomic tetrasomic set (N4AT4D, N4AT4B, N4BT4A, N4BT4D, N4DT4A and N4DT4B). The region spanning *Ms1* was then identified by amplification of these markers on the sterile deletion mutant series *ms1a*, *ms1b* and *ms1c*[15–19] relative to their respective fertile wild-type (*Ms1*) controls[15–19].

An F$_2$-mapping population derived from a cross between the male sterile cv. *Chris*-EMS-mutagenised line FS2 (*ms1d*)[21] and male fertile cv. *Gladius*[46] was

developed. *Ms1*-flanking markers identified by deletion mutant mapping were converted into high-resolution melting (HRM) markers[47] and tested for polymorphism between parental genotypes, and then assayed for segregation within the F$_2$ population. Recombinants within the *Ms1* region were identified from F$_2$- and F$_3$-derived individuals using a combination of both HRM markers and KBioscience competitive allele-specific polymerase chain reaction (KASPar) assays[48] based on SNPs used to develop both the 9K[49] and 90K[50] iSelect Beadchip Assay from Illumina. Primers used for either HRM or KASPar assays are listed in Supplementary Table 5 and 6. All recombinants for the region were marker selected, then fertility tested and not male sterile subsequently progeny tested for fertility in order to determine the zygosity.

**BAC clone analysis.** Eighteen probes were designed within the 0.5-cM region bounded by markers wsnp_Ex_c18318_27140346 (*x27140346*) and wsnp_Ku_c7153_12360198 (x12360198), using synteny to *Brachypodium* and rice. The probes were designed to be non-repetitive based on BLAST analysis of target sequences. The probes were then PCR amplified and separated by agarose gel electrophoresis with fragments of desired size being eluted from the gel using a Qiaquick Gel Extraction kit (Qiagen, Germantown, MD, USA). PCR fragments were pooled to an equimolar concentration and then $^{32}$P-dATP radiolabelled by a NEBlot kit (New England Biolabs) using a manufacturer's protocol. The labelled probe was purified in a Sephadex G50 column (GE Healthcare) and denatured at 100 °C for 10 min. Twenty eight high-density BAC clone colony filters gridded onto Hybond N + nylon membranes (GE Healthcare, Piscataway, NJ, USA) were used for hybridisation. This represents a coverage of 5.1-genome equivalents from the *Triticum turgidum ssp.* durum wheat cv. Langdon[42]. For prehybridisation, overnight incubation of colony filters in a hybridisation solution (2x SSPE, 0.5% SDS, 5x Denhardt's reagent[51] and 40 μg ml$^{-1}$ salmon sperm DNA) was done in rotary glass tubes at 65 °C. The labelled probe was mixed with 5 ml of hybridisation solution and colony filters were incubated at 65 °C overnight. To remove the unbound probe, the filters were washed twice in a washing solution containing 2x SSPE and 0.5% SDS and rinsed with 1x SSC. The washed filters were exposed to an X-ray film for 1–3days based on the signal intensity to identify positive clones. BAC clones that gave a positive signal were grown on single colonies from glycerol stabs and then DNA was extracted according to the BACMAX$^{TM}$ DNA purification kit (Epicentre®, www.epicentre.com, Madison, Wisconsin, USA).

Restriction mapping, PCR experiments with primers corresponding to the markers previously used, determined the order of the BACs covering the region of interest. BAC libraries from the *Triticum turgidum ssp.* durum cv. Langdon and a bread wheat proprietary cultivar were screened with probes from the *Ms1* region, with positive clones being selected for sequencing. All BACs were sequenced using the Illumina MiSeq platform with paired-end (PE) reads of 250 bp. Quality-controlled PE reads were mapped in a single-end mode to the bread wheat cv. Chinese Spring chromosome arm survey sequencing using Biokanga 2.76.2 (https://github.com/csiro-crop-informatics/biokanga) allowing 2 mismatches per 100 bp to confirm that they were derived from homeogroup 4. The reads were then filtered for bacterial sequence contamination, and trimmed for a vector sequence using a combination of BLASTn-based filters and custom scripts. Before assembly, the overlapping PE reads were fused using FLASH 1.2.7 (https://sourceforge.net/projects/flashpage). The fused reads along with the remaining PE reads were then assembled into contigs using MIRA 4.0[52]. Contigs produced by MIRA were then scaffolded using SSPACE[53] using mate-pair anchors for each contig derived from a single mate-pair library for all samples. Single contiguous scaffolds for each homeolocus were manually finished using Gap5[54]. Highly repetitive regions on the BACs were masked based on a per-base depth of mapped reads (cv. Excalibur, cv. Gladius[55]) exceeding 1000. The alignments to the BACs of *Brachypodium* genes as well as mappings of publically available RNA-seq datasets from the bread wheat cv. Chinese Spring[56] facilitated gene prediction. Nucleotide sequences spanning the *Ms1* region were submitted to GenBank (accession codes KX447407, KX447408 and KX447409).

**Nucleic acid extraction and expression analysis.** DNA extractions from all bread wheat lines were performed using either a phenol/chloroform or freeze-dried extraction protocol[57]. A 15 cm leaf piece from a 2-week-old plant was frozen in liquid nitrogen, and the tissue was ground to a fine powder using one large (9 mm) and three small (3 mm) ball bearings and a vortex. 700 μl of the extraction buffer (1% sarkosyl, 100 mM Tris-HCl at pH 8.5, 100 mM NaCl, 10 mM EDTA and 2% PVPP) was added to each sample and the samples were mixed for 20 min on a rotary shaker. 700 μl of phenol/chloroform/iso-amylalcohol (25:24:1) was added and the extract was transferred to a silica matrix tube and spun at 4000 rpm for 10 min. DNA was precipitated by adding 60 μl of 3 M sodium acetate at pH 4.8 and 600 μl of isopropanol and centrifuged at 13 000 rpm for 10 min. The DNA pellet was washed with 1 ml of 70% ethanol, centrifuged for 2 min at 13 000 rpm and air dried for 20 min. The purified DNA was resuspended in 50 μl of R40 (1x TE, 40 μg ml$^{-1}$ RNase A).

*TaMs1* and homeologous transcripts were detected by qRT-PCR on cDNA using total RNA extracted from wheat cv. *Chris* (wild type) using an ISOLATE II *RNA* Mini Kit (Bioline). For the anther developmental series, a single anther per floret was squashed in acetocarmine and mounted for microscopy. Microspores were cytologically examined for the stage of development. The remaining two

anthers from the same floret were isolated and snap frozen in liquid nitrogen. Developmentally equivalent anthers were pooled and RNA isolated. All total RNA samples were treated with DNase I (Qiagen). First-strand cDNA was synthesised using oligo dT[51] and Superscript III reverse transcriptase (Thermo Fisher). Amplification products from qRT-PCR on each tissue sample, three technical replicates and three biological replicates were used to estimate *TaMs1* and the homeologue transcript abundance relative to *TaEFA2*2*, *TaGAPdH 2*2* and *TaCyclophilin 2*2* reference transcripts. Standard qRT-PCR assays[58] were performed using primers, as listed in Supplementary Table 7.

**RNA-seq expression analysis**. RNA-seq reads derived from five organs (root, leaf, stem, spike and grain) at three developmental stages each from hexaploid wheat cv. Chinese Spring have been published previously[56]. The reads were aligned against the repeat-masked BAC assemblies with Bowtie2[59] and Tophat2[60]. The returned alignments were stringently filtered so as to remove ambiguously mapped reads and read pairs with conflicting alignments. Gene expression was computed on RNA-seq data by using Cufflinks and Cuffmerge v.3.0[61]. RNA-seq expression data for all predicted coding regions from the BAC assemblies are presented in Supplementary Table 1.

**TaMs1 sequence from mutant alleles**. To identify the *TaMs1* sequence variants in the *ms1d*, *ms1e* and *ms1f* alleles, the *TaMs1* coding region was amplified from 13 individuals segregating for the sterility phenotype for each mutant allele. PCR used Phusion High-Fidelity DNA Polymerase (NEB, M0530) with Phusion GC buffer, 5% DMSO and 1 M betaine using the primer *TaMs1* (coding region) listed in Supplementary Table 7. The fragments were subcloned into a pCR™8/GW/TOPO® TA Cloning Kit (Thermo Fisher Scientific, K250020) and Sanger sequencing of positive clones was performed by the Australian Genome Research Facility. Sequence chromatograms were compared using Geneious version 6.1.8[62]. Sequence analysis correlated a G-to-A transition at position 329 with the sterility phenotype in the *ms1d* mutant. A Kompetitive Allele-Specific PCR (KASP™) (LGC Genomics) was designed to this SNP transition using Primer Picker (LGC Genomics) (007-0091.1, Supplementary Table 6) and it was assayed using KASP Mastermix on the SNPline (LGC Genomics) on DNA from the *ms1d* x *Gladius* F$_2$-mapping population, the *ms1* mutant alleles and across a panel of 192 spring wheat germplasm.

**Phylogenetic analysis**. *TaMs1* homologous Poaceae sequences were retrieved from Phytozome (www.phytozome.net), TGAC *Triticum monococcum* Shotgun sequence, International Barley Genome Sequencing Consortium and Rice Genome Annotation Project (http://rice.plantbiology.msu.edu). All BLASTn, BLASTp, tBLASTn and BLASTx hits were retrieved using a cutoff e-value of ≤ 1 × 10$^{-5}$. Default BLAST settings were used for querying with complete sequences. Two prediction tools, PredGPI (gpcr.biocomp.unibo.it/predgpi/) and big-PI Plant Predictor (mendel.imp.ac.at/gpi/plant_server.html), were then used to determine whether primary peptide sequences contained a putative GPI-anchored motif at the C-termini. Protein multiple-sequence alignments (MSAs) were generated using MUSCLE (default settings) implemented in Geneious analysis package (www.geneious.com). Manual alignment was performed to improve the MSAs. The phylogenetic tree was computed with MEGA7 using the maximum likelihood method under default parameters (www.megasoftware.net).

**Constructs**. A 4.3-Kb genomic fragment containing approximately 1.5 Kb upstream of *TaMs1* start codon and 1 Kb downstream of the stop codon was synthesised and introduced upstream of the visible marker MoPAT-DsRED (a translational fusion of the bialaphos resistance gene, phosphinothricin-N-acetyl-transferase, and the red fluorescent protein DsRED) transcribed by the maize Ubiquitin promoter[63] and this 8.1 kb DNA fragment replaced the 2.1 kb *Hind III-Eco RI* DNA fragment from PHP43534[64]. This plasmid was introduced into *A. tumefaciens* strain LBA4404[65] by electroporation using a Gene Pulser II (Bio-Rad) [66] and used for complementing the *ms1d* mutant. A transcriptional reporter construct containing 1.5 Kb of *TaMs1* promoter sequence was fused to *gusplus*[67] and subcloned into the binary vector pMBC32.

**Transformation and in vitro culture**. Male-sterile transformation-amenable spring wheat lines, which contain the *ms1d* allele, were developed by backcrossing with pollen from the spring wheat-transformable cv. *Gladius*. Either immature embryos that segregated for the *ms1d* allele (3:1) or the cv. *Fielder* were used as donor materials for transformation. cv. *Fielder* donor material was used to test the transcriptional fusion construct with the *gusplus* reporter gene. Here, *A. tumefaciens* strain ALG0 (pBGXI) was engineered to contain a disarmed pTiBo542 carrying the *TaMs1::gusplus* cassette in a pMBC32 backbone. Segregating *ms1d* donor material enabled the transformation directly into a homozygous *ms1d* background to test for genetic complementation of the *ms1d* mutation with the putative *Ms1* allele in the transformed plants (T$_0$). To generate wheat transformants[68] for testing either complementation of the *ms1* mutation or the *TaMs1* transcriptional reporter, wheat (*Triticum aestivum* L., cv. *Gladius* or cv. *Fielder*) plants were grown in a growth chamber at 18/15 °C (day/night), with a 16-hr photoperiod (minimum of 1000 μmol s$^{-1}$ m$^{-2}$ light) or in a greenhouse. Immature seeds with immature embryos (IEs) of about 1.5–2.5 mm collected from spikes 12–14 days post anthesis were surface-sterilised for 20 min in 15% (v v$^{-1}$) bleach (5.25% sodium hypochlorite) plus one drop of Tween 20 followed by three washes in sterile water. IEs

were isolated and placed in 1.0 ml of liquid infection medium (WI4; MS salt + vitamins (4.43 g l$^{-1}$), maltose (30 g l$^{-1}$), glucose (10 g l$^{-1}$), MES (1.95 g l$^{-1}$), 2,4-D (1 ml, 0.5 mg l$^{-1}$), Picloram (200 μl, 10 mg ml$^{-1}$) and BAP (0.5 ml, 1 mg l$^{-1}$)) with 0.25 ml of autoclaved sand into 2 ml microcentrifuge tubes. IEs were treated by centrifuging at various strengths in an infection medium and then inoculated with *Agrobacterium*. The suspension of *Agrobacterium* and IEs was poured into a co-cultivation medium (a WI4 medium containing 5.0 μM CuSO$_4$ without glucose solidified with 3.5 g l$^{-1}$ of Phytagel). IEs were then placed on an embryo axis side down on the media, and incubated in the dark at 21 °C. After three days, IEs were transferred to a DBC4 medium containing 100 mg l$^{-1}$ of cefotaxime (Phyto-Technology Lab., Shawnee Mission, KS) and then incubated at 26–28 °C under dim light for two weeks. The DBC4 medium is a DBC3 green-regenerative medium[69] modified with 1.0 mg l$^{-1}$ of 6-benzylaminopurine (BAP). The tissues were then transferred to a DBC6 medium (a modified DBC3 medium with 0.5 mg l$^{-1}$ of 2,4-dichlorophenoxyacetic acid and 2.0 mg l$^{-1}$ of BAP) containing 150 mg l$^{-1}$ of cefotaxime for another two weeks. Regenerable *DsRed*- expressing transgenic sectors were identified using a Leica M165 FC fluorescence microscope, cut from the non-transformed tissues and placed on a MSA regeneration medium [MSB[68] without indole-3-butyric acid] with 150 mg l$^{-1}$ of cefotaxime, whereas *TaMs1::gusplus*-containing tissues were selected based on resistance to 100 mg l$^{-1}$ of hygromycin B. After sectors have developed into small plantlets, they were transferred to an MSB-rooting medium. During each transfer, plantlets were checked for *DsRed* gene expression and any non-expressing or chimeric tissues were removed.

**T$_0$ plantlet generation and analysis**. T$_0$ wheat regenerants containing a single- or multicopy *TaMs1–DsRed* cassette(s) were identified by copy number qPCR[70] using the following forward, reverse and probe primers (Fwd: 5′-GACATCCCCGAC-TACAAGAAGCT-3′, Rev: 5′-CACGCGCTCCCACTTGA-3′ and Probe1-FAM MGB 5′-CCTTCCCCGAGGGC-3). Zygosity was shown of the *ms1d* mutation using flanking and linked markers (Supplementary Table 5 and 6). Plantlets were transferred to the glasshouse and assessed for self-fertility and expression of *DsRed* fluorescence in the resulting seed. Seed counts from these individual plants were counted as a qualitative measure of male fertility.

**Molecular and phenotypic traits of T$_1$ plants**. Inheritance of complementation by *TaMs1* T-DNA insertion was shown by analysing a selfed seed set on T$_1$ plants derived from two separate T$_0$ plants, each with independent T-DNA insertions (Event-1 and Event-7) (Supplementary Table 4). One set of T$_1$ progenies was derived from a T$_0$ plant homozygous for *ms1d* mutation (*ms1d/ms1d*) and containing the *TaMs1–DsRed* cassette (Event-1). The second set of T$_1$ progenies was derived from a T$_0$ plant heterozygous for *ms1d* mutation (*Ms1/ms1d*) and containing the *TaMs1–DsRed* cassette (Event-7). Genotyping for *ms1d* zygosity and the presence of the T-DNA insertion for plants derived from both sets were determined using flanking markers, as described above, and the expression of the *DsRed* seed colour marker.

**Data availability**. MiSeq BAC-sequencing data have been deposited in the NCBI SRA database under Bioproject ID PRJNA396428. The assembled genomic DNA of BACs derived from the *Ms1* locus (KX447407) as well as its A (KX447408) and D (KX447409) genome-derived homeoloci have been deposited in NCBI GenBank. Further data that support the findings of this study are available from the corresponding author upon reasonable request.

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

## Acknowledgements

We thank J. Hayes for critical discussions and reading of the manuscript, P. Warner, M. Ovitigala, S. Manning, A. Okada, Y. Li, and M. Kumar for technical assistance. We also thank C. Kastner and M.-J. Cho for transgenic line generation, D. Dias and U. Roessner for metabolomic profiling and G. Mayo for microscopy assistance. This research was

funded by DuPont Pioneer Hi-Bred International Inc. We are grateful for the support provided by The University of Adelaide, Australian Research Council, Grains Research and Development Corporation and the South Australian State Government.

## Author contributions

E.J.T., U.B., A.K., M.B., T.O., M.G., C.D., Y.W., A.S., P.L., P.W., M.C.A., A.M.C. and R.W.: Designed the experiments. E.J.T., A.K., M.B., T.O., C.D., R.S., A.S. and M.S.: Performed the experiments. E.J.T., U.B., A.K., C.D., R.S., A.M.C., and R.W.: Analysed the data. E.J.T., P.L., A.M.C., M.C.A., and R.W.: Wrote the manuscript.

## Additional information

**Competing interests:** M.A., U.B., M.C., M.S., E.T. and R.W. have filed a patent on this work. The remaining authors declare no competing financial interests.

