## [Peer Review File · Nature Communications]

Reviewers' comments:

Reviewer #1 (Remarks to the Author):

This paper identified a nuclear encoded recessive male sterile gene, *Tams1*, encoding a glycosylphosphatidylinositol anchored lipid transfer protein using map-based cloning and functional complementation in wheat. Authors analyzed the function domain of LTPGs using mutations induced by EMS. Finally, authors discussed the application value of *Tams1* in hybrid breeding. This is not easy to clone gene from wheat using map-based cloning. This work is also interesting to me. However, the characterizations of *Tams1* and *Tams1* mutations are incomplete and the application of *Tams1* in this paper is just speculative lacking of experimental evidence.

Major comments

1. This manuscript is a letter not article. In my impression, Nature Communications only publishes the article. This manuscript should be rewritten according to request.
2. The title is not suitable. Authors only cloned the *Tams1* and proposed the application method of *Tams1* in hybrid breeding. However, it is impossible that only functional complementation plants are applied in practical hybrid breeding. Because it is very difficult to isolate a large number of non-transgenic seeds from functional complementation plants on a large scale. An entire SPT system needs to contain (1) a complementary wild-type male fertility gene to restore fertility, (2) an α -amylase gene to disrupt pollination and (3) a seed colour marker gene.
3. The part that authors tested the ability of *TAMS1* gene sequence to complement *ms1d* should be a part of functional complementation.
4. The characterizations of *Tams1* mutations are insufficient. Authors should supplement the detail morphological defects using the semi-thin and ultrathin section analysis of anthers in WT and mutants. PCD of tapetum should be analyzed using TUNEL. I advise authors to detect the wax and cutin in WT and mutation for function analysis of LTPGs.
5. Authors should construct phylogenetic trees of LTPGs to understand the evolutionary and functional conservation of LTPGs. This is important to detect subcellular localization of *Tams1* for its function analysis.

Minor comments

1. Figure 2b
The ordinate scale is not appropriate.
2. Please supplement the pollen sterility picture of different *ms1* mutant.

Reviewer #2 (Remarks to the Author):

The authors report cloning of *Ms1* gene for hybrid wheat seed production as a part of the genetic male sterility based system as opposed to currently used cytoplasmic sterility-based system for hybrid wheat production. This is a major breakthrough for an alternative system for hybrid wheat production which may enhance yield by 10% or more. Some of my

comments are directly written on the marked copy of the ms. I have hard time getting used to the gene being called Ms1 as it was always known as ms1. The authors may be aware that symbol Ms is used for dominant male sterile (Ms) mutant series located on chromosomes 4D and 5A. Those are also mal sterile mutants and they should be discussed in the context of hybrid seed production. I suggest that authors consult McIntosh's catalog of Gene Symbols about the gene designation. I also indicated on the ms that authors extremely negative comments about the CMS system although that is the only system that is currently used in commerce. The data relating to the cloning of the gene and its validation is solid but discussion is weak. I was possible to produce hybrid wheat using ms1 by several systems, how the availability of the gene makes this process more efficient? Hybrid seed production scheme using gene sequences? All known mutants are located on chromosome 4B and one one (ms5) on 3A? Why no mutants have been recovered on 4A or 4D? Is the Ms3 mapped on 4D by Chinese scientists homoeologous?

bikram gill

Reviewer #1 (Remarks to the Author):

This paper identified a nuclear encoded recessive male sterile gene, *Tams1*, encoding a glycosylphosphatidylinositol anchored lipid transfer protein using map-based cloning and functional complementation in wheat. Authors analyzed the function domain of LTPGs using mutations induced by EMS. Finally, authors discussed the application value of *Tams1* in hybrid breeding. This is not easy to clone gene from wheat using map-based cloning. This work is also interesting to me. However, the characterizations of *Tams1* and *Tams1* mutations are incomplete and the application of *Tams1* in this paper is just speculative lacking of experimental evidence.

Major comments

1. This manuscript is a letter not article. In my impression, Nature Communications only publishes the article. This manuscript should be rewritten according to request.

The manuscript has been restructured as an article as suggested.

2. The title is not suitable. Authors only cloned the *Tams1* and proposed the application method of *Tams1* in hybrid breeding. However, it is impossible that only functional complementation plants are applied in practical hybrid breeding. Because it is very difficult to isolate a large number of non-transgenic seeds from functional complementation plants on a large scale. An entire SPT system needs to contain (1) a complementary wild-type male fertility gene to restore fertility, (2) an α -amylase gene to disrupt pollination and (3) a seed colour marker gene.

As suggested, the tone of the title has been changed to better reflect the content of the manuscript, given we do recognise that additional genetic elements to the complementing gene sequence are required to establish a commercial hybrid breeding platform such as SPT for wheat. The revised manuscript also attempts to place more focus on what was the key challenge of identifying and cloning a recessive male sterility gene from wheat. We do however, provide additional evidence showing α -amylase and seed colour marker functionality in wheat (Supplementary Fig. 9) plus a schematic depicting how the dominant *TaMs1* male fertility restoring gene sequence, when coupled with these two elements, is used within a SPT hybrid breeding scheme (Supplementary Fig. 1). We believe this additional evidence and clarity now better reflects the contents of the manuscript.

3. The part that authors tested the ability of *TaMS1* gene sequence to complement *ms1d* should be a part of functional complementation.

Taking the reviewers concern into consideration the manuscript has been revised as per point 2. The text has been articulated to place additional emphasis on complementation being the remaining key component for SPT hybridization platform development when considering the additional evidence for α -amylase and seed colour marker functionality (Supplementary Fig. 9).

4. The characterizations of *Tams1* mutations are insufficient. Authors should supplement the detail morphological defects using the semi-thin and ultrathin section analysis of anthers in WT and mutants. PCD of tapetum should be analyzed using TUNEL. I advise authors to detect the wax and cutin in WT and mutation for function analysis of LTPGs.

Additional evidence is provided that *TaMs1* functions specifically in sporopollenin biosynthesis or transport (Supplementary Fig. 3-5). Light microscopy revealed *ms1* defects are first observed in early

uninucleate microspores (Supplementary Fig. 3) with TEM microscopy on ultrathin sections (Supplementary Fig. 4) also reveal reduced electron dense materials at the tapetal cell surface in *ms1d* versus wild-type (*Ms1*) anthers, both in line with a role for *TaMs1* in sporopollenin precursor transport between the tapetal cells and the developing microspores. Additionally, GC-MS (Supplementary Fig. 6) on mutant compared to wild-type anthers revealed an increase in lipid monomers of sporopollenin, C16 and C18 long chain fatty acids. These findings provide further evidence that *TaMs1* disruption inhibits sporopollenin biosynthesis or transport. These seem specific to tapetal cell/microspore exine development considering no differences were observed in the epidermal wax/cuticular layer structure between mutant and wild-type anthers (Supplementary Fig. 5). Taken together, we believe these findings do not necessitate a TUNEL assay for assessing programmed cell death in tapetal cells.

5. Authors should construct phylogenetic trees of LTPGs to understand the evolutionary and functional conservation of LTPGs. This is important to detect subcellular localization of *Tams1* for its function analysis.

A phylogenetic tree showing that *TaMs1* is a member of a *Poaceae* specific clade of LTPGs is presented in Supplementary Fig. 2.

Minor comments

1. Figure 2b

The ordinate scale is not appropriate.

If we were to remove the log scale in the chart as per the reviewer's suggestion then a true comparison between *TaMs1* and its homeologues is difficult to draw. We therefore believe that keeping the log scale prevents a biased interpretation of *TaMs1*'s sub-genome specific expression.

2. Please supplement the pollen sterility picture of different *ms1* mutant.

The pollen sterility phenotypes as indicated by acetocarmine-glycerin staining for wheat male sterile mutants *ms1d* (FS2), *ms1e* (FS3), *ms1f* (FS24) have been previously described by Sasakuma *et al.*, 1978. In addition to what has previously been reported, we provide Supplementary Fig. 8 as it compiles bright field images and average pollen viability counts as indicated by acetocarmine and Lugol staining for *ms1d*, *ms1e*, *ms1f* and *ms1h* relative to wild type (*Ms1*) controls when grown under our glasshouse conditions.

Reviewer #2 (Remarks to the Author):

The authors report cloning of *Ms1* gene for hybrid wheat seed production as a part of the genetic male sterility based system as opposed to currently used cytoplasmic sterility-based system for hybrid wheat production. This is a major breakthrough for an alternative system for hybrid wheat production which may enhance yield by 10% or more. Some of my comments are directly written on

the marked copy of the ms. I have hard time getting used to the gene being called Ms1 as it was always known as ms1. The authors may be aware that symbol Ms is used for dominant male sterile (Ms) mutant series located on chromosomes 4D and 5A. Those are also male sterile mutants and they should be discussed in the context of hybrid seed production.

I suggest that authors consult McIntosh's catalog of Gene Symbols about the gene designation.

We did indeed consult McIntosh's catalog of Gene Symbols from 2004-2017 but found the nomenclature for male sterile mutant loci to be somewhat inconsistent. For example, chromosomal male sterile mutant loci have been designated ms (ie. ms1, 2, 3, 4, 5), which as listed (IWGS, 2012) does not specifically take into account the recessive or dominant nature of their respective mutant loci (ie *ms1* on chr. 4BS is recessive whilst *Ms2* on chr. 4DS is dominant). This seems to be inconsistent with additional detail provided on the dominance/recessive nature of mutants in McIntosh's annual Gene Symbol supplements (eg. recessive *ms1g* in the 2007 supplement and dominant *Ms1376* in the 2012 supplement). Accordingly, we chose to provide as much information in the manuscript on the type of mutant loci being investigated. In this case *ms1* (and its alleles *ms1a, b, c, d, e, f* and *h*) specifically refers to the recessive mutant locus of ms1 (McIntosh's nomenclature) on chr. 4BS, whilst *Ms1* refers to the cognate dominant wild-type fertility restorer to this recessive mutant locus. Upon gene sequence identification of this dominant fertility restorer, we chose to term it *TaMs1*, with *Ta* designating an abbreviation for *Triticum aestivum*, therefore preventing confusion with many other *Ms1* gene sequences derived from model organisms (ie. *Arabidopsis, maize*).

I also indicated on the ms that authors extremely negative comments about the CMS system although that is the only system that is currently used in commerce.

We believe the reviewer has misinterpreted the statement "wheat male sterility and restoration systems" to refer specifically to CMS-based systems, however this was not the intention since we are actually referring to systems like CHAs and several PTGMS systems.

We recognise that CMS is currently used in commerce, but reports highlight that its use has been severely inhibited by the need to introgress and track multiple restorer and modifier loci in both the male and female pools. This complexity has led to the high costs of hybrid seed production with these systems (refer to Cisar & Cooper, 2002). The advent of marker assisted or genomic selection using tightly linked molecular markers to major and minor restorer loci (Ma & Sorrells, 1995) may help alleviate these problems and support more extensive CMS deployment.

Taking the reviewers concern into consideration we have modified the statement to "Wheat male sterility and restoration systems were first developed in the 1960s, but many proved impractical and commercially high risk"

The data relating to the cloning of the gene and its validation is solid but discussion is weak. I was possible to produce hybrid wheat using ms1 by several systems, how the availability of the gene makes this process more efficient? Hybrid seed production scheme using gene sequences?

We refer to reviewers 1's second point whereby we addressed a similar concern by providing a schematic of the SPT breeding scheme (Supplementary Fig. 1). This specifically depicts how the *TaMs1* gene sequence, when linked on the same transgenic construct as a seed colour marker and pollen germinator inhibitor, can be used in a maintainer line for bulking male sterile female inbred seed. We have also elaborated within the text to describe how efficient bulking of male sterile (*ms1/ms1*) female inbred seed is a cost saving measure for hybrid seed production. Furthermore, we

point out, that knowledge of both the mutant and wild-type fertility restorer gene sequences allows efficient introgression, tracking and selection of sterility and fertility restoration. How the SPT hybridisation platform overcomes weaknesses inherent to the 4E-*ms* cytogenetic system is also discussed.

All known mutants are located on chromosome 4B and one one (*ms5*) on 3A? Why no mutants have been recovered on 4A or 4D?

The reviewer's question on why no mutants have been recovered on chr. 4A or 4D is important. Reports indicate that two dominant male sterile mutants (*Ms2* and *Ms4*) have been identified on chr 4DS (see following question below), however no recessive male sterile mutants have been located on these chromosomes. With a view to characterising recessive *ms1*, we also asked this question. However reports of fertility in nullisomic-tetrasomic (Sears, 1953; Sears 1964, Sears and Sears 1978), and deletion sets (Endo, 1991; Endo & Gill, 1996) for group 4 chromosomes (http://www.k-state.edu/wgrc/genetic_resources/deletion_lines/group_4.html), show that complete 4B chromosome loss or 4BS telomere loss (bin 4BS1-0.81-1.00) induces complete male sterility (Sears and Sear 1978; the chromosome is designated 4A in the literature)(Endo, 1991; Endo & Gill, 1996). This compares to lines where chromosomes 4A or 4D are lost which retain some fertility. It should be noted that terminal deletions of pericentric chromosome 4A, 4AL-3 and 8, which are the most proximal deletions of this chromosome, were also reported to have reduced seed set. However, there was no clear evidence that this was a consequence of incomplete male sterility, but may have resulted from meiotic chromosomal instability (or asynapsis), since these involved large terminal deletions. Furthermore, deletion mutant bin mapping (Qi et al., 2004) shows that *TaMs1* A and D homoeologues are located in deletion bins 4AL13-0.59-0.66 and 4DS2-0.82-1.00 respectively. Homozygous deletion mutants used to define these bins were not reported as completely male sterile, nor needed to be maintained as heterozygotes considering they are available as fixed homozygote lines (ie. 4AL-12, 13 and 5, at http://www.k-state.edu/wgrc/genetic_resources/deletion_lines/group_4.html). Taken together, this data would indicate that in the cv. Chinese Spring, chr. 4BS is necessary for male fertility, with little affect from chromosomes 4A and 4D. It should be noted that this affect appears to be genotype dependent considering a self-fertile ditelosomic line for the long arm of chromosome 4B was recently created using cv. Norin 61 (Joshi et al., 2013). In addition, on a molecular level, our qRT-PCR experiments show that the *TaMs1* 4A, 4B and 4D homeologues are not co-expressed temporally (Figure 2b) further indicating that there is a functional divergence of the homoeologues at the expression level.

Is the *Ms3* mapped on 4D by Chinese scientists homoeologous?

Dominant male sterile mutation *Ms3* has been mapped to chr. 5AS (Qi & Gil, 2001). We believe the reviewer may have been referring to *Ms2* (synonymous with 'Ta1' or 'Taigu') which to our knowledge is one of two dominant male steriles reported to be located on chr 4DS. The other is *Ms4* (Maan & Kianin, 2001). The *Ms2* locus has been investigated by Chinese and Canadian scientists (Deng & Gao 1982, Ji & Deng 1985, Liu & Deng 1986, Deng 1988, Sui & Sun, 2001, Cao et al., 2009). We have no evidence that dominant *Ms2* is homoeologous to the recessive *ms1* considering *Ms2* mapped proximal to *RhtD1c* in two independent studies (ie. 0.18 cM in Yang et al, 2009 and 1 cM in Cao, 2009), whereas mapping in our bi-parental populations of both *TaMs1* and its D homoeologue revealed that they were both distal to *RhtB1* and *RhtD1* respectively. For example, in a *ms1d* (Chris) x Pastor population (n=307) *TaMs1* mapped 18.7 cM distal to *RhtB1*, whereas *TaMs1*'s D homoeologue mapped 25 cM distal to *RhtD1* in a Cornerstone (*ms1c*) x Gladius population (n=155). Observations of *ms1* versus *Ms2* anthers also showed their phenotypes are very different to one another. Moreover, the very recent cloning and complementation of the *Ms2* gene sequence *WMS*,

reveals that it is not a homeologue of *TaMs1*, but instead represents a novel gene sequence expressed only in sterile anthers as a consequence of a transposon insertion (refer to patent WO/2016/193798 and Ni *et al.*, 2017). As a discussion point we have modified the text to incorporate this information. *Ms4* was deemed to be located on chr. 4DS via ditelo mapping (Maan & Kianin, 2001), so further research would also be needed to determine whether this mutant locus is homoeologous to *ms1*.

One of the most cost effective and flexible hybridization platforms that uses a recessive male sterile is Seed Production Technology (SPT)³ developed for maize and rice hybrid seed production.

Comment: “my information is that this platform has been abandoned and does not represent a major advantage. Of course we need many systems of hybrid production and marketplace will decide which one is most successful.”

The maize SPT hybridisation process has been de-regulated and implemented commercially in maize for the USA (APHIS DP-32138-1) as it increases production efficiency and reduces costs (https://www.aphis.usda.gov/brs/aphisdocs/08_33801p.pdf). We agree with Reviewer 2's point that the marketplace will ultimately determine which hybridization platform is the most successful. However, this does not negate that SPT currently sits as one of the most cost effective and flexible hybridization platforms based on a recessive nuclear encoded male sterile.

For example, only ten wheat male-sterile mutants have been identified to date⁸, in contrast to 108 in diploid barley^{10,11}. Polyploidy not only makes it difficult to find suitable male sterile mutations but also complicates deploying mutants since multiple mutations would be needed to deal with genic redundancy¹²

Comment: “It is really not true, wheat has diploid genetics as most phenotypic traits give diploid inheritance. yes polyploidy is a complicating factor but only when you get into molecular mapping.”

Although polyploid wheat behaves as a diploid in inheritance this does not negate that there are usually three homoeologous sequences that can be redundant in function. Frequently mutant phenotypes are only observable in wheat if multiple homoeologues are mutated. It is only in cases where there is functional divergence between the homoeologues that a single or double mutant can induce a mutant phenotype. The identification of so few recessive male sterile wheat lines through mutagenesis relative to the diploid barley, indicates that there is a high level of genetic redundancy on a homoeologue and a genome scale. The identification of *TaMs1* as a single locus controlling fertility in polyploid wheat is quite remarkable. The deployment of this mutant in a commercial hybrid breeding platform, such as SPT, will be cost effective since there is only one mutation to maintain, one locus to track and one complementing sequence required.

REVIEWERS' COMMENTS:

Reviewer #1 (Remarks to the Author):

The author's response basically meets my remarks. I find the manuscript was significantly improved in this revised version. However, authors should discuss the reason of increased long chain fatty acids in ms1d anthers. Why all kinds of long chain fatty acids are accumulated especially C16 and C18 long chain fatty acids in ms1d anthers?

Two independent groups cloned ms2 in wheat, but this manuscript only cited a reference.

Reviewer #2 (Remarks to the Author):

I have read the reviewed copy of the ms. Obviously, authors have done considerable amount of work since the last review and have thoroughly addressed reviewer's comments, certainly this reviewer's. Fig. 1 proposing the use of the gene sequence in hybrid wheat platform is very elegant and all components are feasible. I have minor comment; the phylogenetic analysis detected close affinity with the 4A locus of *T. monococcum*, how about any hits in *Aegilops speltoides*- where the orthologous copy may be found?

Reviewer #1:

The author's response basically meets my remarks. I find the manuscript was significantly improved in this revised version. However, authors should discuss the reason of increased long chain fatty acids in ms1d anthers. Why all kinds of long chain fatty acids are accumulated especially C16 and C18 long chain fatty acids in ms1d anthers?

We have revised the manuscript to include discussion on the reason for increased C16 and C18 long chain fatty acids (see lines 201-208).

Two independent groups cloned ms2 in wheat, but this manuscript only cited a reference.

A second reference to the recent cloning of *Ms2* was included in the manuscript. Xia, C. *et al.* A TRIM insertion in the promoter of *Ms2* causes male sterility in wheat. *Nature Communications* 8, 15407 (2017).

Reviewer #2:

I have read the reviewed copy of the ms. Obviously, authors have done considerable amount of work since the last review and have thoroughly addressed reviewer's comments, certainly this reviewer. Fig. 1 proposing the use of the gene sequence in hybrid wheat platform is very elegant and all components are feasible. I have minor comment; the phylogenetic analysis detected close affinity with the 4A locus of *T. monococcum*, how about any hits in *Aegilops speltoides*- where the orthologous copy may be found?

The quality of the *Aegilops speltoides* genome assembly (TGAC WGS *speltoides* V1) currently does not permit the retrieval of a *TaMs1* orthologous sequence.